# The neuronal chaperone proSAAS is highly expressed in the retina

**Nicholas Schaffer**[1], **Samira Mitias**[1], **Yan Guo**[2], **Steven L. Bernstein**[2], **Iris Lindberg**[1]*

**1** Department of Neurobiology, University of Maryland School of Medicine, Baltimore, Maryland, United States of America, **2** Department of Ophthalmology, University of Maryland School of Medicine, Baltimore, Maryland, United States of America

* ilindberg@som.umaryland.edu, ilind001@umaryland.edu

## Abstract

The many layers of the neuroretina contain a complex, interconnected network of specialized neurons that both process visual stimuli and conduct processed information to higher brain areas. Neural networks rely on proteostatic control mechanisms to maintain proper protein homeostasis both in cell bodies as well as within synapses; protein chaperones play an important role in regulating and supporting this process. ProSAAS is a small neuronal chaperone that functions as an anti-aggregant in *in vitro* assays and is released upon depolarization in neuronal primary cultures. We here report a potential role for proSAAS in the retina. A review of human and mouse retinal RNAseq studies reveals that proSAAS expression is abundant within the retina. Single cell sequencing data from mouse and human studies show that proSAAS levels are highest in retinal ganglion cells (RGCs) and horizontal cells. Using proSAAS antibodies in combination with antisera to known retinal cell markers in mouse retinal sections, we confirm RNAseq data showing that proSAAS expression is highest in RGCs and horizontal cells. The proSAAS signal is concentrated within the ganglion cell layer and the inner plexiform layer, a dense synaptic layer connecting retinal neurons. Western blotting of mouse retinal extracts indicates the presence of two processed proSAAS forms, a 21 kDa C-terminally processed form, and a small 13 kDa species which, based on antibody specificity, likely represents an internal fragment. This fragment is also found in extracts prepared from human retinas. Taken together, our data provide support for the hypothesis that retinal synapses utilize the proSAAS chaperone to support visual signaling.

## Introduction

The neuroretina is responsible for the conversion of light signals into processed nerve impulses that are transmitted to the brain via the axons of retinal ganglion cells (RGCs), whose cell soma reside in the inner retina. Normal retinal activity requires precise protein homeostasis to maintain normal functionality [1,2]. Proteostatic dysregulation is a disorder in which misfolded proteins contribute to cellular diseases.

**Data availability statement:** All relevant data are within the paper and its Supporting Information files.

**Funding:** NIA AG062222 to IL R01 EY032519 to SLB The funders had no role in study design, data collection and analysis, decision to publish, or preparation of the manuscript.

**Competing interests:** The authors have declared that no competing interests exist.

**Abbreviations:** A-II, A-II Amacrine cells; A-G, GABA-Amacrine cells; A-G, Glycinergic Amacrine cells; C-BPf, OFF-Cone bipolar cell neurons; C-BPo, ON -Cone bipolar neurons; R-BP, Rod bipolar neurons; RGC, retinal ganglion cells.

Proteostatic dysregulation is a major component in age-related vision decline [3,4], as retinal neurons become progressively impaired in their ability to properly manage protein turnover [5]. Disrupted proteostasis has also been linked to retinal dystrophies [reviewed in [2]], as well as to glaucoma, a pathology with concurrent proteostatic and synaptic dysfunction coupled to progressive RGC degeneration [6].

Molecular chaperones are crucial in maintaining proper protein homeostasis in neuronal cells. Multiple molecular chaperones are upregulated to relieve the cellular stress associated with protein misfolding [reviewed in [7]]. These chaperones include both heat shock proteins as well as chaperones connected with lysosomal and autophagic processes [7]. However, few neuronal chaperones are known to handle extracellular aberrant proteostatic stressors.

The proSAAS chaperone (gene name *PCSK1N;* chromosome location *Xp11.23*) is highly expressed in brain [8,9] and in synapses [10], from which it is secreted [11]. ProSAAS has been shown to block the fibrillation and aggregation of both Abeta 1–42 and alpha synuclein [reviewed in [12]]; to relieve endoplasmic reticulum stress [13,14]; and to prevent the death of nigral neurons when virally expressed in a synuclein overexpression model of Parkinson's disease [15], supporting a cytoprotective function. Interestingly, while beta amyloid plaques are present in the aging human brain, and are especially prominent in human Alzheimer's disease (AD) and in mouse AD models, there is relatively little amyloid plaque development in the human and mouse retina [16–18], despite high concentrations of soluble amyloid precursor protein [19]; reviewed in Wang *et al* [20]]. These data suggest the existence of possible chaperone mechanisms that might block intraretinal plaque formation.

The proSAAS chaperone has been associated with synaptic homeostasis [11,15], and proSAAS overexpression has been shown to reduce plaque burden in an AD model mouse [11]. However, to date no analysis of proSAAS expression and distribution within the retina has been performed. In the current report, we evaluated proSAAS expression, location, and molecular forms in the mouse and human retina.

## Materials and methods

### *PCSK1N* Meta-analysis

We accessed ARCHS4, a publicly available repository of RNA-Seq data for aligned human and mouse sequences [21]. A total of 2738 samples of human retinal samples across 79 batches and 3045 mouse samples across 155 batches were filtered for the expression of three transcripts: clusterin, proSAAS, and α-crystallinβ. Expression data for each species was log(2) + 1 transformed and then normalized for batch effects using the ComBat algorithm [22]. A total of 2738 samples of human retina were filtered for the expression of three transcripts: clusterin, proSAAS, and α-crystallinβ.

### Analysis of *PCSK1N* expression from human and mouse single Cell RNAseq studies

We compared cellular retinal *PCSK1N* gene expression in depth using data generated by two single cell RNAseq studies, one human and one mouse [23,24] using the Broad Institute database website: https://singlecell.broadinstitute.org/single_cell [25].

## Animals

All procedures involving live animals and euthanasia were approved by the Institutional Animal Care and Use Committee (IACUC). Tissues from C57BL/6J mice of mixed gender were harvested only from animals following euthanasia (decapitation after isoflurane anesthesia). Tissues were kept on ice until retinal and hippocampal dissection; post-mortem interval ranged from 1–6 h. The two hippocampi and retinas obtained from each animal were pooled.

## Human tissue

Retinas were obtained from a single human donor (male, age 30) harvested on 10/20/2016, pooled, and frozen at -80C; the post-mortem interval is unknown. Tissue collection was approved under the UMB IRB exemption issued by the University of Maryland-Baltimore; studies on human tissue followed the Declaration of Helsinki guidelines.

## Acetic acid extraction of retinas and hippocampi

Mouse retinal and hippocampal proSAAS was extracted with acetic acid as per our previous publication [26]. Dissected retinas and hippocampi were placed in 0.5 ml of 1 M ice-cold acetic acid and immediately manually homogenized using pellet pestles (Sigma-Aldrich; St. Louis, MO; Cat#: Z359971). Samples were then sonicated to reduce viscosity and ensure homogenization, flash frozen on dry ice, then thawed on wet ice. Aliquots were taken for measurement of total protein content using the Pierce BCA assay (ThermoFisher; Waltham, MA; Cat#: 23225), and the remaining samples were centrifuged at 14,000 x g for 10 min at 4°C to pellet insoluble material. The clear supernatants were frozen and lyophilized overnight. The dried acid extracts were resuspended in Laemmli sample buffer (50 mM Tris-HCl, pH 6.8, 6 M urea, 10% glycerol, 2% SDS, 5% beta mercaptoethanol, and 0.8% Bromophenol blue) and stored at -20°C.

Human retinas were homogenized in 1 ml of ice-cold extraction buffer (20 mM Tris-HCl, pH 8.0, 100 mM NaCl, 0.5 mM EDTA and 0.5% NP-40, supplemented with phosphatase inhibitors (Roche; Basel, Switzerland; Cat#: 04906845001) and protease inhibitors (ThermoFisher; Cat#: A32955). Following manual homogenization, samples were sonicated to reduce viscosity and a sample was taken for protein determination by the BCA method. An aliquot of the remaining homogenate was diluted into Laemmli sample buffer containing 6 M urea. Samples (9% of the total retinal homogenate; 12 µg of protein) were loaded onto 16% Mini Invitrogen Novex Tris-Glycine gels (ThermoFisher) and Western blotting for proSAAS was performed as described below.

## Western blotting

For proSAAS blotting of mouse hippocampal and retinal samples, 16% Mini Invitrogen Novex WedgeWell Tris-Glycine gels (ThermoFisher) were used together with the PageRuler Plus ladder (ThermoFisher; Cat#: 26619). Acid extracts, resuspended in Laemmli sample buffer, were heated to 90°C for 5 min, and an amount corresponding to approximately 160 µg of original homogenate protein was loaded (representing 8% of the total retinal protein and 33% total hippocampal protein); we note that the retinal sample contained considerable acid-insoluble protein, thus the final protein load was reduced in retina comparison to the hippocampus). Gels were electrophoresed at 150–200 volts and then transferred to 0.2 µm nitrocellulose membranes using a Bio Rad Trans-Blot Turbo semi-dry transfer apparatus (Bio-Rad; Hercules, CA; Cat#: 1704270) for 12 min at 1.3 amps and 24 volts. Residual gel proteins were stained using Coomassie (25% methanol, 10% acetic acid, 0.0003% Brilliant Blue) in order to validate protein loading consistency and to assess transfer efficiency. Following transfer, membranes were washed briefly in phosphate-buffered saline (PBS) and cross-linked with 1% glutaraldehyde (Sigma-Aldrich; Cat#: G5882) in PBS for 20 min. Glutaraldehyde fixation has been shown to greatly increase proSAAS immunoreactivity (proSAAS-ir) and to reduce background [13,27]. Following glutaraldehyde fixation, membranes were rinsed with Tris-Buffered Saline containing 0.05% Tween-20 (TBST) and then incubated in blocking solution (5% milk in TBST) (Bio-Rad; Cat#: 1706404) for 30 min. Primary antisera were diluted in blocking solution and incubation in the

respective primary antibody occurred overnight at 4°C with gentle rocking. Fig S1 depicts the uncropped Western blots and residual Coomassie gels showing protein loading.

Two different proSAAS primary antibodies were used for Western blotting: polyclonal rabbit IgG (LS46) generated against recombinant His-tagged 21 kDa mouse proSAAS (residues 1–180) [28], and a commercial mouse monoclonal pro-SAAS antibody generated against a full-length GST-tagged human proSAAS (1E9; Millipore-Sigma cat.# SAB1402570; RRID: AB_10637837). These antibodies were both used at a dilution of 1:1000.

Regarding antibody specificity, the LS46 antiserum has been shown to react by Western blotting with recombinant 21 and 27 kDa proSAAS and cDNA-encoded forms of proSAAS, as well as with the tissue equivalents of these forms [11,13]. The LS46 antiserum also reacts with an endogenous 17 kDa proSAAS fragment [13] which is present in, and released from, hippocampal synapses in culture upon depolarization [11]. The proSAAS 1E9 antiserum was obtained from Millipore-Sigma, and was generated in rabbits against a GST-conjugated, proSAAS internal peptide: DGPAGPDAEE-AGDETPDVDPELLRYLLGRILAGSADSEGVAAPRRLRRAADHDVGSELPPEGVLGALLRVKRLETPAPQVPARRLLPP. Company Western blot data of proSAAS-transfected HEK cells show that this antiserum reacts with a single band of intact proSAAS which does not appear in untransfected cells. Neural and endocrine cells, known to proteolytically process pro-SAAS [8,26] were not tested. Our data show that this commercial antiserum has similar reactivity to LS46 in reacting with full-length 27 kDa, 21 kDa, 17 kDa and 13 kDa forms of proSAAS (**Fig 6** and unpublished data).

Following reaction with primary antibodies, membranes were washed with TBST and incubated at room temperature for 2 h in horseradish peroxidase (HRP)-conjugated secondary antisera (1:5000 in blocking solution). Secondary antisera included HRP-linked anti-rabbit antiserum (Jackson ImmunoResearch Labs; West Grove, PA; RRID: AB_2307391) and HRP-linked anti-mouse antiserum (Bio-Rad; RRID: AB_11125547). Following additional rinses in TBST, membranes were developed using Bio-Rad Clarity ECL substrate (Bio-Rad; Cat#: 1705061) and imaged on a Bio Rad ChemiDoc.

## Cryosectioning

Intact mouse eyes harvested from postmortem mice were punctured in the sclera with a 27-gauge needle, then placed in 4% paraformaldehyde (PFA) overnight at 4°C. Following fixation, eyes were moved to 30%[1] sucrose for at least 24 h prior to removal of the lens and cornea. The remaining eye cup was then flash frozen on dry ice in Optimal Cutting Temperature compound (OCT; Sakura; Torrance, CA; Cat #: 4583). Ten-micron sections were cut using a Leica cryostat 3050S and affixed to positively-charged microscope slides (Fisher Scientific; Hampton, NH; Cat#: 12-550-15); slides were stored at 4°C prior to immunofluorescence experiments.

## Immunofluorescence

Mounted retinal cross-sections, prepared as described above, were bordered in 100% silicone gel and rehydrated in PBS + 0.05% Tween-20 (PBST) for 5 min. Slides were incubated in blocking solution (2% normal donkey serum; Jackson Immuno Research; RRID: AB_2336990, in PBST) for 1 h at room temperature, then incubated overnight at 4°C in primary antisera diluted in blocking solution. Primary antisera included the LS46 proSAAS antiserum described above; and antisera to choline acetyltransferase (ChAT; 1:500; Millipore; Burlington, MA; RRID: AB_94647), calbindin (1:500; Sigma-Aldrich; RRID: AB_476894); Brn3a (1:500; Synaptic Systems; Göttingen, Germany; RRID: AB_2737037), glial fibrillary acidic protein (GFAP; 1:500; Millipore; RRID: AB_211868); IIIβ-tubulin (1;500; Sigma-Aldrich; RRID: AB_1844090), RBPMS (RNA Binding Protein with Multiple Splicing; 1:500, GeneTex; Irvine, CA; RRID: AB_10720427); melanopsin antiserum (1:500; MilliporeSigma; Burlington, MA; Cat#: MABN770). A goat proSAAS antiserum directed against human proSAAS, residues 182–193 (78% identical with mouse proSAAS in this region) (Abcepta; San Diego, California, Cat#: ALS16157; used at 1: 1000) was used for the multiple staining experiment shown in Fig 3B and 5B-C. At least 3 random fields were imaged for each marker gene discussed. Slices were selected only for layer integrity, not for any other properties.

Post-primary antibody incubation, slides were washed 3x in PBST, then incubated in fluorophore-conjugated secondary antibodies diluted in PBST for 2 h at room temperature. Secondary antisera used for immunofluorescence included: Alexa Fluor™ 647 donkey anti-mouse (1:500; Invitrogen; RRID: AB_162542), Cy3 Donkey anti-rabbit (1:500; Jackson ImmunoResearch Labs; RRID: AB_2307443), Cy2 Donkey anti-goat (1:500; Jackson ImmunoResearch Labs; RRID: AB_2340420), Cy2 Donkey anti-rat (1:500; Jackson ImmunoResearch Labs; RRID: AB_2340673), and Alexa Fluor™ 647 Goat anti-guinea pig (1:500; Invitrogen; RRID: AB_2735091). Following additional washing with PBST, slices were counter-probed with DAPI nuclear stain in PBST for 15 min, washed again, and mounted in fluorescence mounting media. Slides were imaged on a Leica SP8 confocal microscope at 20x magnification with 1024 x 1024 pixel resolution. Z-stacks capturing the entire slice were taken and merged as maximal projections, with brightness levels adjusted to increase contrast and eliminate background fluorescence. No-primary antibody controls were included in each experiment to monitor the degree of non-specific staining; Fig S1 depicts the no-primary antibody control images for Fig 3A and Fig 4.

### Immunofluorescence quantitation

Raw maximum projected images from immunofluorescence experiments conducted as described above were quantitatively assessed for proSAAS immunoreactivity using Fiji v1.54k. To assess proSAAS content in retinal layers, regions of interest (ROIs) for each of the GCL, IPL, INL, OPL, ONL, and PR layers (GCL = Ganglion Cell Layer, IPL = Inner Plexiform Layer, INL = Inner Nuclear layer, OPL = Outer Plexiform Layer, ONL = Outer Nuclear Layer, PR = Photoreceptor Layer) and of the non-retinal background were manually selected and measured for corrected total fluorescence (CTF) with the following equation: CTF = Integrated Density- (ROI area x background average) in both experimental and no primary control images. To account for individual experimental variation and autofluorescence, CTF values were normalized (divided) by the CTF obtained from no-primary control images in the same experiment.

To assess proSAAS-ir content in cell marker-positive cells, minimum cross entropy thresholding [29] was implemented in Fiji to generate ROIs around cell marker-positive regions above background via background subtraction of a 50 pixel rolling ball. Since some cell types span multiple retinal layers, the mean fluorescence of all layers (except the PR layer, due to high autofluorescence) was measured and added to the background of each image to produce a "cumulative background" value used to calculate the CTF.

To quantify the proSAAS-ir content of various RGCs, minimum cross entropy thresholding was implemented to select melanopsin-positive and Brn3a-positive cells following background subtraction. Since comparisons were made among observations within the same experiment, proSAAS mean fluorescence was measured, rather than CTF, in Brn3a-positive and in melanopsin-positive cells. Melanopsin ROIs were subtracted from Brn3a ROIs to quantify Brn3a-positive and melanopsin-negative RGCs. Either unpaired t-tests, or two-way ANOVA with Tukey's post-tests were used to determine statistical significance, as noted in the figure legends.

Fig S1 depicts the no-primary antibody control images for Fig 3A and Fig 4.

## Results

### Meta-analysis of transcriptomic studies shows that proSAAS is expressed in multiple cell types in the retina

To assess the presence of proSAAS (*PCSK1N/Pcsk1n*) as a component of the normal human and mouse retina transcriptome, we accessed ARCHS4, a massive publicly available repository of RNAseq data [21]. We compared normalized proSAAS transcript counts across all studies to those of two other known retinal extracellular chaperones: clusterin and α-crystallin-β, small chaperones associated with retinal proteostasis and anti-aggregation [30,31]. ARCHS4 human and mouse datasets obtained under different experimental conditions were log-transformed and normalized with the ComBat algorithm, which uses an empirical Bayes model to normalize batch effect-derived variability. Retinal proSAAS transcript expression was robust, confirming that proSAAS is a component of the normal human retinal transcriptome, but was lower than that of clusterin and α-crystallin-β in humans (**Fig 1A**). Clusterin was found at high levels in both human and mouse

retina, consistent with previously reported findings [30,31]. Interestingly, in mouse retinal samples, *Pcsk1n* expression was found at higher average levels than α−crystallinβ, and exhibited a wider distribution among studies, possibly as a result of a larger number of batches (**Fig 1B**). Overall, these data show that proSAAS is widely expressed in both human and mouse retinal transcriptomics studies.

In order to compare proSAAS expression in the various types of human and mouse retinal cells, two single cell RNA-seq studies [23,24] were analyzed using Broad Institute software. This analysis showed that relative proSAAS expression is greatest in horizontal cells in both species, followed closely by RGCs and amacrine cells (Fig2A-F). UMAP projections of single cell sequencing data show low to medium *PCSK1N/Pcsk1n* transcript quantities that are widespread across cell clusters (**Fig 2B**-C, 2E-F). Interestingly, Muller glia in both species express proSAAS, albeit at a lower level than other cell types (**Fig 2D**, F).

We then analyzed murine proSAAS expression in individual RGC subtypes, using single cell RNASeq data from the Broad Institute database employing RGC subtype-specific cell clustering [30] (**Fig 2G**). Inter-study variability in single cell data is likely due to sequencing batch effects, which should not impact comparisons of intra-study transcript quantities between clusters. While we identified robust proSAAS expression across all RGCs, in both studies, proSAAS transcripts were clearly more abundant in melanopsin-1-associated RGCs ("M1"). These cells represent intrinsically photoreceptive RGCs (ipRGCs) that contribute to circadian rhythm signaling, though not to vision processing [32–34].

To determine if retinal proSAAS expression is associated with proteostatic diseases with visual manifestations, we performed a literature search of -*omics* studies using accessible online datasets containing associations with proteostasis. We found five such studies across *Homo sapiens* and *Mus musculus* datasets which identified retinal proSAAS [3,35–38]. Of these, three studies found a statistically significant increase in proSAAS in retinal diseases linked to proteostatic dysfunction (glaucoma, age-related macular degeneration (ARMD), and retinitis pigmentosa, while two did not, including a single AD study (**Fig 2H**). Glaucoma is a disease of RGCs and their corresponding axons, which comprise the optic nerve. While the glaucoma proteomics data support disease–related upregulation of proSAAS levels, additional studies of proSAAS levels in retinal disorders involving proteostatic dysregulation are needed to confirm this positive association.

## Retinal proSAAS is enriched in the GCL and IPL retinal layers and synapses, and is present in optic nerve tissue

To validate results obtained from the transcriptional meta-analysis, we experimentally characterized proSAAS distribution in mouse retinal tissue using cross-sections immunostained for proSAAS and retinal cell markers. We determined that proSAAS-immunoreactivity (proSAAS-ir) is strongest surrounding the RGC somatic/cytoplasmic marker RBPMS (ribonu-cleic acid binding protein with multiple splicing) [39] (**Fig 3A**). Quantitation of proSAAS-ir in retinal layers showed diffuse proSAAS-ir present above background in all layers; the strongest immunoreactive regions were the RGC/Ganglion cell layer (GCL) and the inner plexiform layer (IPL), which spans the area between the GCL and Inner Nuclear Layer (INL) and is made up of RGC dendrites and synaptic junctions between RGC and amacrine neurons (**Fig 3B**) [40]. We further evaluated IPL proSAAS expression by co-localizing the neuron-specific marker IIIβ-tubulin with proSAAS-ir, revealing that proSAAS-ir is enriched in IPL-associated neuronal processes (**Fig 3C**). These data support substantial proSAAS presence in retinal synapses. ProSAAS-ir was also found in the optic nerve head, but the signal was weaker than that seen in the synaptic bundles themselves (**Fig 3C**).

## Retinal proSAAS is expressed in RGCs, amacrine, and horizontal cells

We co-immunostained mouse retinal cross-sections with antisera against both proSAAS and choline acetyltransferase (ChAT), a marker for cholinergic amacrine cells and their synaptic endings in the IPL sublayers. This revealed substantial proSAAS-ir signal surrounding both normal and displaced ChAT-positive putative amacrine cells in the IPL and GCL layers. ProSAAS-ir was found in the ChAT-positive synaptic layers of both ON and OFF amacrine cells [40] in the IPL (**Fig 4A**). ProSAAS co-localization also occurred with the astrocyte marker glial fibrillary acidic protein (GFAP) (**Fig 4B**),

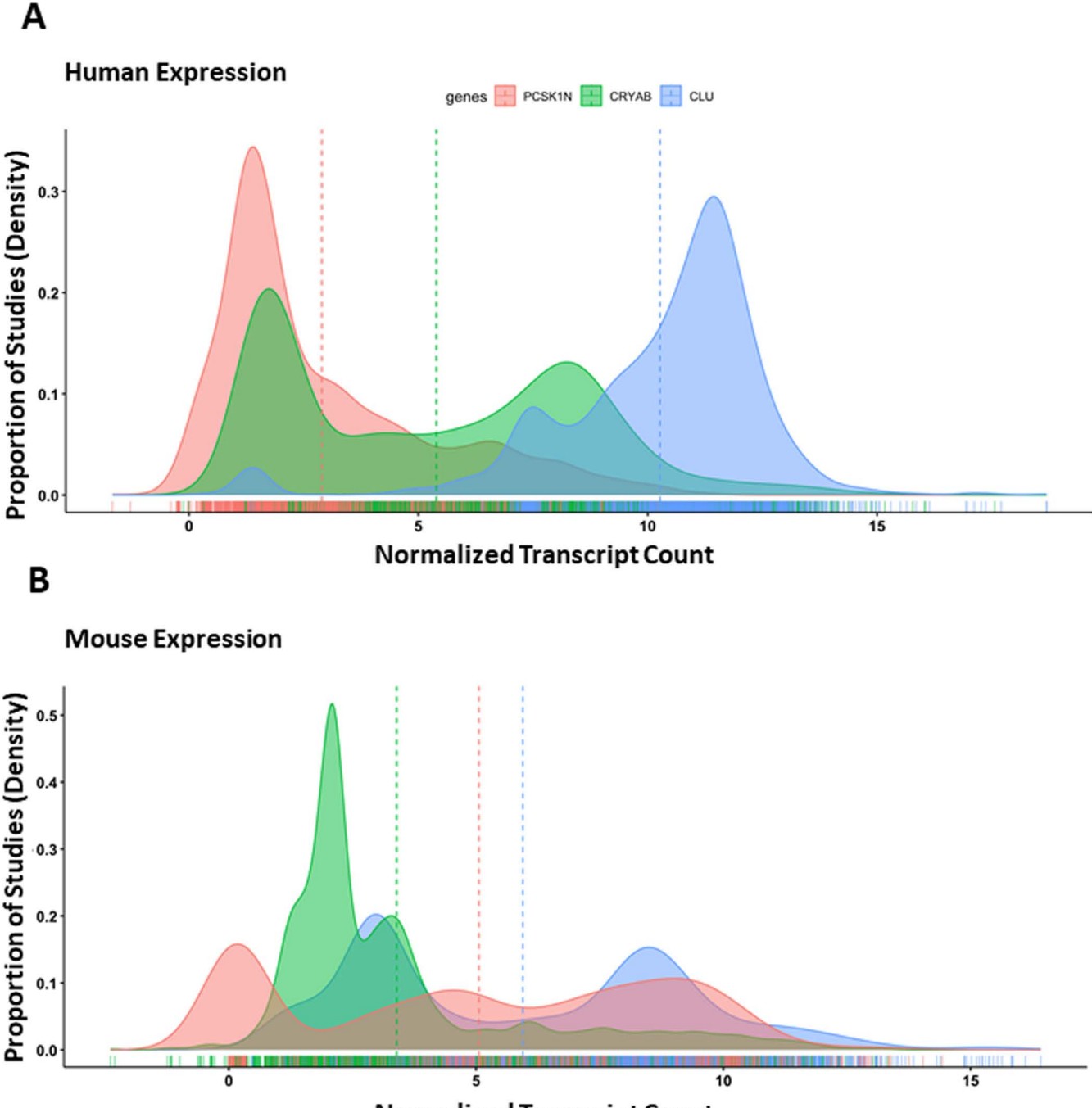

**Fig 1. ProSAAS is expressed in multiple cell types in a mixed species meta-analysis.** Log-transformed transcriptional counts of proSAAS (*Red*); clusterin (CLU, *Blue*); and αcrystallin-β (CRYAB, *Green*) transcripts in human and mouse studies obtained from ARCHS4, were normalized using the ComBat batch correction algorithm. Expression distributions are represented as a continuous density function, with vertical lines representing the means of each distribution. **Panel A**: Human: n = 2737 samples, 79 batches; **Panel B:** Mouse: n = 3045 samples, 155 batches).

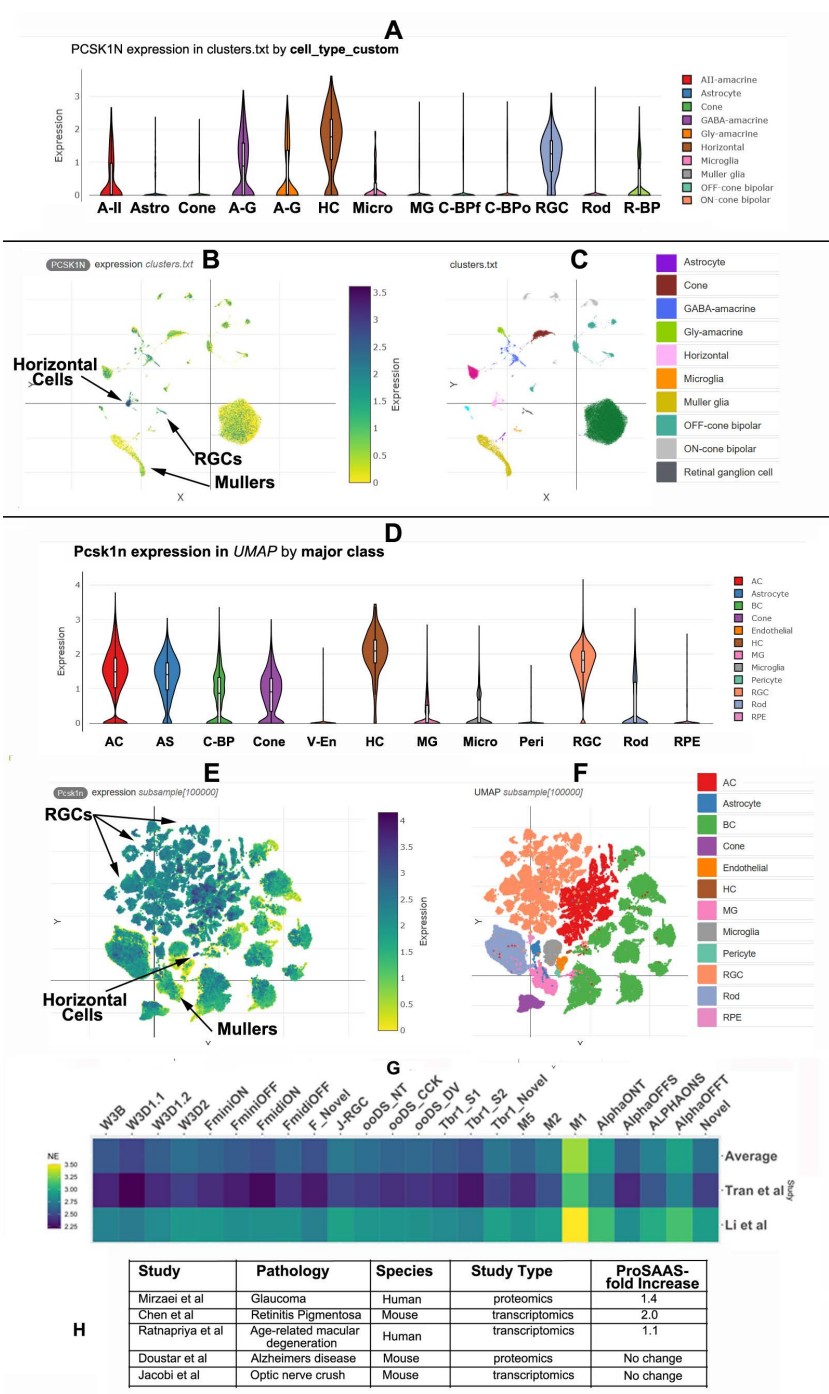

**Fig 2. ProSAAS is expressed in multiple cell types in the human and rodent retina.** proSAAS transcript expression in retinal cell types of: **Panels A-C: Human; Panels D-F: Mouse. Panels A, B**: Differential cell expression plots. **Panels B-C** and **E-F:** UMAP projections and differential cell expression plots. Specific cell types are indicated by *arrows* in **panels B** and **E**, and by *color* in **panels C** and **F**. Relative proSAAS transcript expression is greatest in horizontal cells in both species, followed by RGCs. RGC subgroup expression is identifiable in the mouse data (seen as orange in **panel F**). Amacrine cell interneurons are subdivided into different types in the human study (**panel A**), and as a single cell type in the mouse study (**panel D**). **Panel G** depicts a comparison of relative *Pcsk1n* mRNA expression in different RGC subtypes [data from [24,30]], while **Panel H** shows a comparison of changes in retinal *PCSK1N* expression in five disorders affecting the retina.

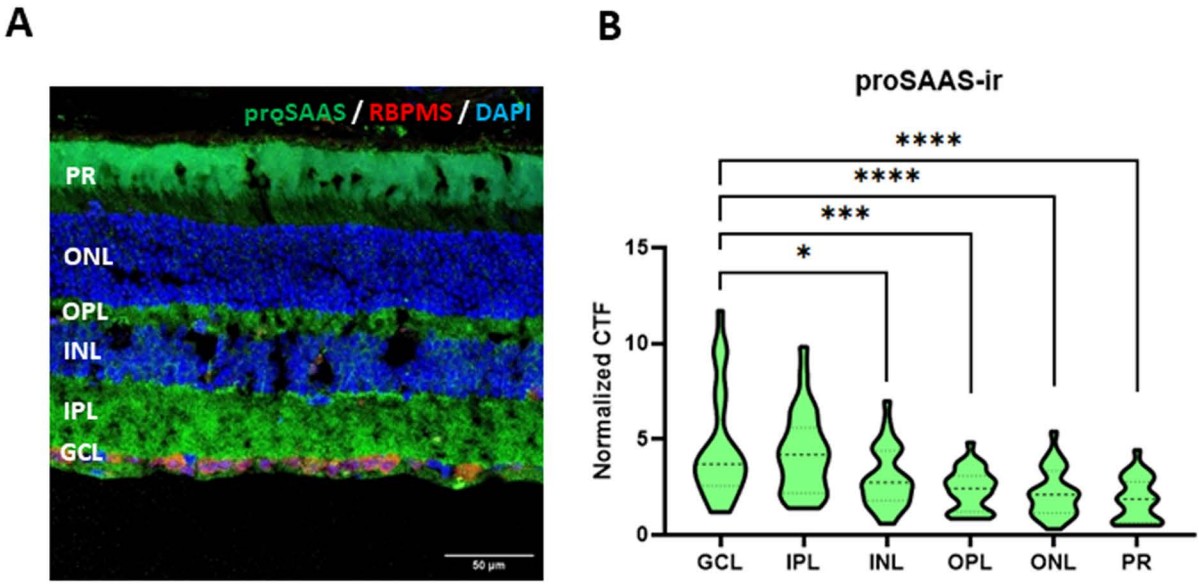

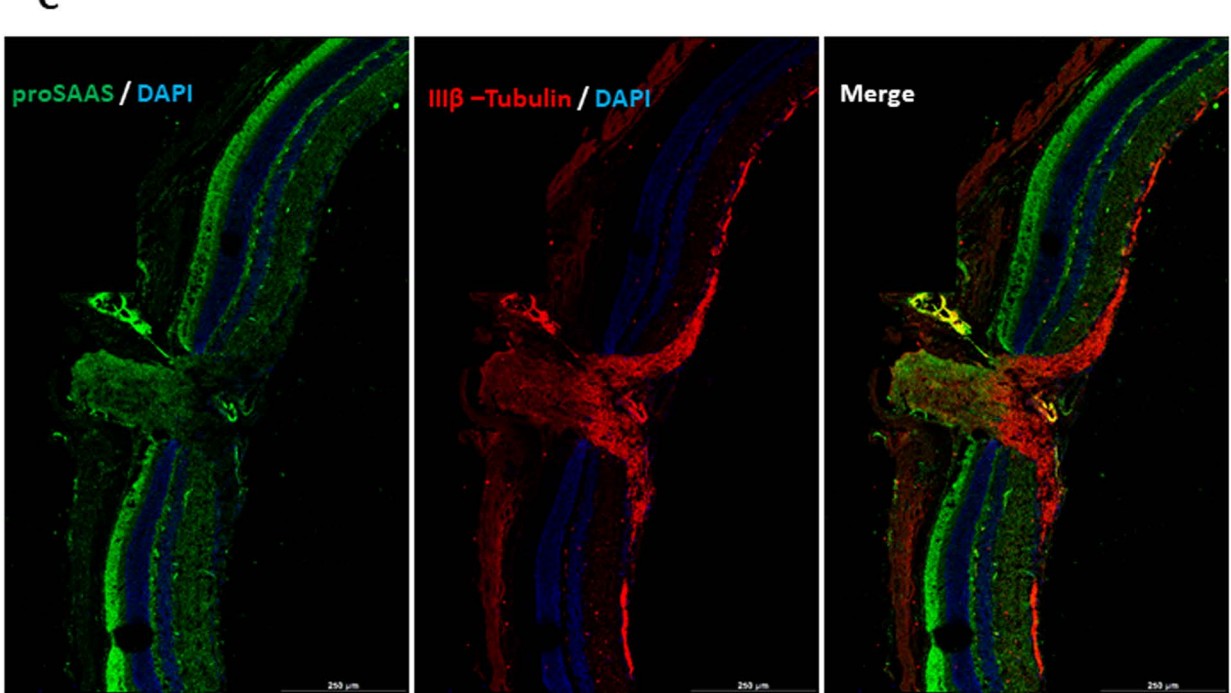

**Fig 3. ProSAAS is enriched in the GCL and IPL mouse retinal layers and is present in synapses and retinal nerve tissue. Panel A**: ProSAAS-ir (LS46 staining) and RBPMS-ir co-staining showing proSAAS-ir surrounding RGCs. **Panel B**: Quantitative analysis of proSAAS-ir in each retinal layer, derived from 28 total images of retinal slices (n = 22 for LS46 antiserum-stained images; n = 6 for proSAAS ALS16157 antiserum-stained images; the choice of proSAAS antiserum was dictated by the available marker primary antibodies. Data are reported as corrected total fluorescence (CTF) normalized to the proSAAS-ir of the no-primary control in each experiment. **Panel C**: Stitched tilescan image of the optic nerve, optic nerve head, and surrounding retinal layers, depicting proSAAS LS46 and IIIβ-tubulin immunostaining. (* p < 0.05, *** p < 0.0001, **** p < 0.00001, unpaired t-test). (GCL = Ganglion Cell Layer, IPL = Inner Plexiform Layer, INL = Inner Nuclear layer, OPL = Outer Plexiform Layer, ONL = Outer Nuclear Layer, PR = Photoreceptor Layer).

despite little transcript expression in astrocytes in our meta-analysis (**Fig 2**). Additional immunostaining with the horizontal cell marker calbindin indicated high levels of proSAAS-ir in horizontal cells as well as in Brn3a-positive putative RGCs, confirming the meta-analysis result (Fig 4C-D).

To quantify levels of proSAAS-ir, co-immunostained images were analyzed for proSAAS-ir as described. Corrected total fluorescence (CTF) was measured for proSAAS in cell marker positive cells. Background and nonspecific signals were subtracted from the regions analyzed (Supplementary Fig 2A-G). This analysis revealed that, as previously shown, proSAAS is significantly enriched in RGCs and horizontal cells. However, in contrast to the transcriptomic data, this quantitation indicated only low levels of proSAAS-ir in putative ChAT-positive amacrine cell bodies. This quantitation also confirmed the presence of proSAAS-ir in GFAP-positive putative astrocytes, though at low levels (**Fig 5A**).

### ProSAAS is present in intrinsically photoreceptive retinal ganglion cells (ipRGCs)

Since proSAAS-derived peptides have previously been associated with circadian rhythm signaling [41], we assessed whether proSAAS is more highly expressed in ipRGCs by co-staining for proSAAS, melanopsin, and Brn3a. A commercial goat proSAAS antibody (Abcepta ALS16157) was used in this experiment for antiserum compatibility. The proSAAS antibody ALS16157 is directed against a short proSAAS sequence, residues 182–193, near the carboxy-terminal portion of proSAAS, and shows a slightly different proSAAS distribution than proSAAS antiserum LS46 (**Fig 5**). ALS16157 proSAAS staining showed proSAAS-ir in melanopsin-positive RGCs (**Fig 5B**), but quantitation of melanopsin-negative/Brn3a-positive and melanopsin-positive/Brn3a-positive RGCs indicated that proSAAS-ir was not preferentially expressed in putative ipRGCs (**Fig 5C**). Thus, in contrast to the scRNAseq data (**Fig 2G**), enhanced proSAAS expression was not detected in melanopsin-containing ipRGCs.

### ProSAAS is differentially processed in the retina and hippocampus

ProSAAS contains four proprotein convertase cleavage sites (**Fig 6A**), many of which are cleaved in neuronal tissues and cell lines [8,26]. We determined the extent of proSAAS processing in mouse retina and hippocampus using Western blotting of retinal and hippocampal protein extracts; we used hippocampus for comparison since it is known to be rich in proSAAS [8,11].

Western blotting revealed robust proSAAS immunoreactivity in the mouse retina, with major bands appearing at 13 kDa and 21 kDa. These bands likely represent a fully processed internal region and the N-terminal 21 kDa domain respectively (**Fig 6B**) [26]. Interestingly, the major proSAAS-ir low molecular mass species in the retina was 13 kDa rather than 17 kDa, as in the hippocampus, indicating possible tissue-specific processing (**Fig 6B**).

Because a 13 kDa proSAAS-ir species has not been previously documented, we performed epitope mapping to confirm that it represents a true proSAAS form. We blotted mouse retinal extracts using both our LS46 proSAAS antiserum (directed against the entire 21 kDa recombinant form; see (**Fig 6A**) as well as the commercial proSAAS antibody 1E9 (directed against the carboxy-terminal third of proSAAS; see **Materials and Methods** and **Fig 6A**). Both antibodies reacted with 21 kDa and 13 kDa forms of proSAAS, albeit with different sensitivities (**Fig 6C**). The fact that two different proSAAS antisera detected the 13 kDa proSAAS form in retinal samples supports the presence of both epitopes, as well the physical presence of this form (**Fig 6C**).

We then investigated the profile of proSAAS-immunoreactive species present in human retinal extracts. Western blotting revealed that the same 13 kDa proSAAS fragment found in mouse retinal extracts is also found in human retinal extracts, supporting similar posttranslational processing (**Fig 6D**).

### Discussion

RGCs process refined visual information obtained from other retinal layers, modified via various interneurons such as bipolar and amacrine cells, and transmit this information to the brain. Retinal neurons, and in particular RGCs, incur a

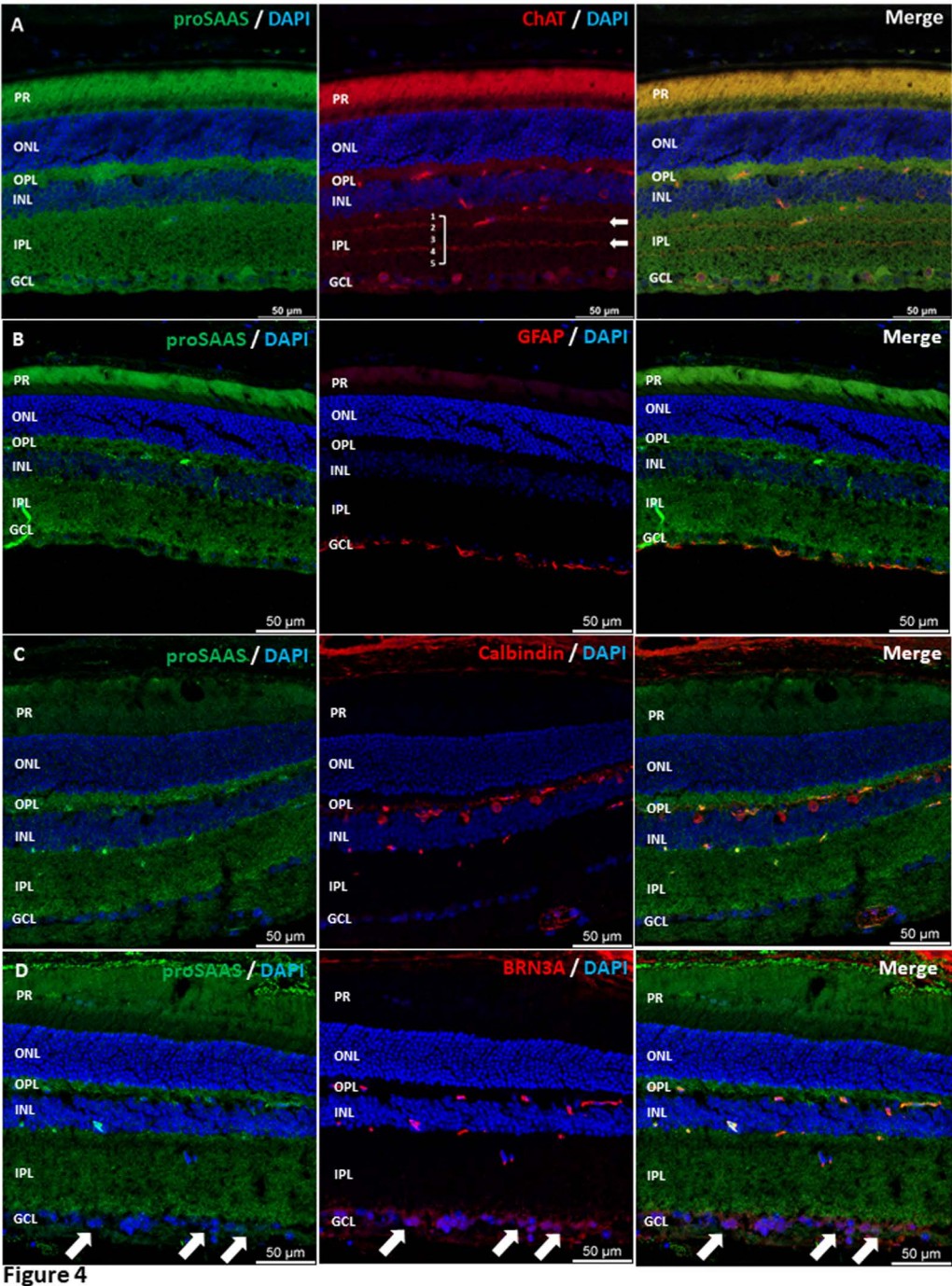

**Figure 4**

**Fig 4. ProSAAS is found in RGCs, amacrine, and horizontal cells in the mouse retina. Panel A:** ProSAAS and ChAT co-staining indicates that proSAAS is in both normal and displaced ChAT-positive amacrine cells, as well as in ChAT-positive synaptic layers in the IPL. **Panel B**: ProSAAS and GFAP co-staining depicts limited proSAAS staining in astrocytes. **Panel C:** ProSAAS and calbindin co-staining shows that proSAAS-ir is enriched in the cell bodies of calbindin-positive horizontal cells. **Panel D:** ProSAAS and Brn3a co-staining depicts proSAAS in and surrounding Brn3a-positive retinal ganglion cells. **Panels A-D:** All images represent maximum projection micrographs of fixed, cross-sectioned retina probed with proSAAS and cell-type marker antibodies. Brightness levels were manually adjusted for each experiment to increase contrast. LS46 proSAAS antiserum was used in this experiment.

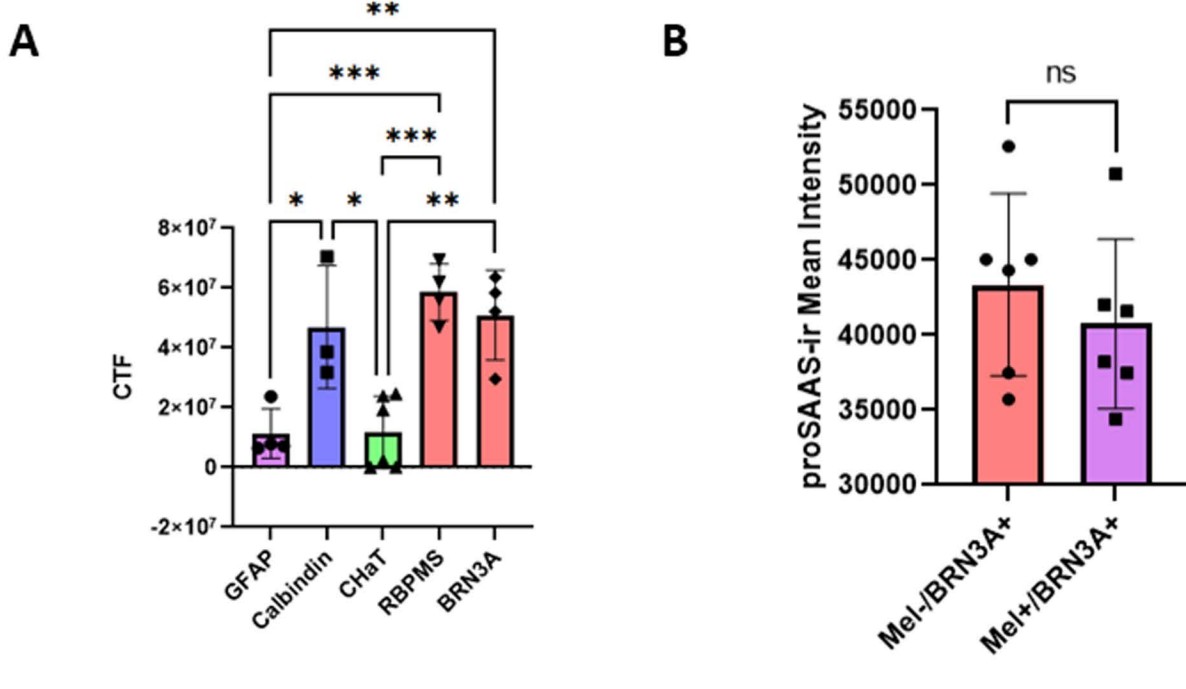

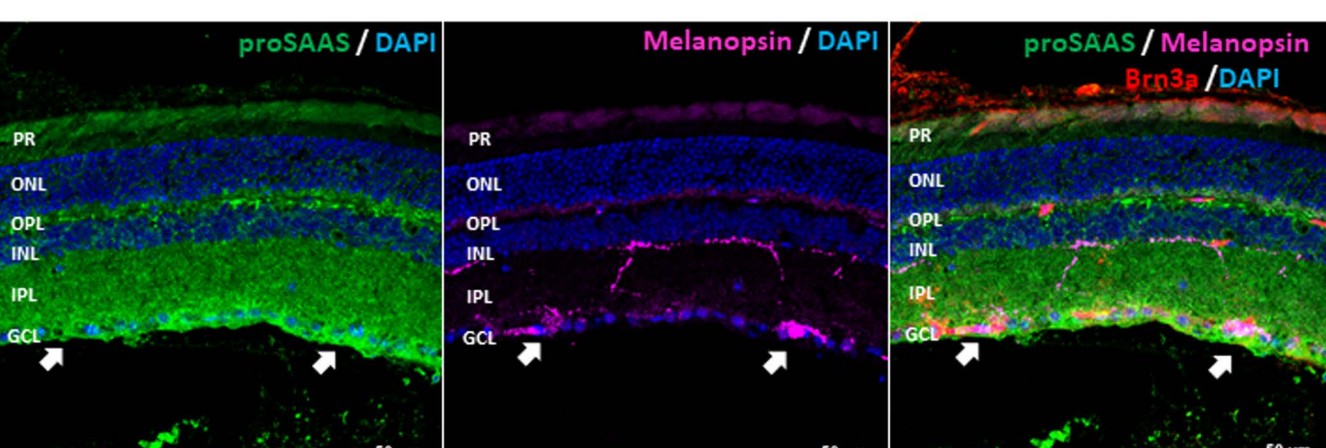

**Fig 5. Image quantitation of RGCs reveals differences in mouse proSAAS distribution as compared to bioinformatics analysis. Panel A:** Quantitative image analysis of retinal cross sections labeled for proSAAS (LS46 antiserum) in cell-marker positive cells (GFAP, n = 4; calbindin, n = 3; CHaT, n = 6; RBPMS n = 4; Brn3a n = 4). **Panel B:** Quantitative image analysis of 6 retinal cross-sections, colabeled with proSAAS, Brn3a, and melanopsin antisera, showing no significant differences. **Panel C:** Representative image of proSAAS, melanopsin, and Brn3a staining. (ns = not significant, * p < 0.05, ** p < 0.01, *** p < 0.0001, **** p < 0.00001, using one-way ANOVA with Tukey's post-test for Panel A; and unpaired Student's t-test for Panel **B.**). For antiserum compatibility purposes, Abcepta ALS16157 goat anti-proSAAS antibody was used for the experiments in Panels **B** and **C.**

large proteostatic load due to their need for continuous adaptation to stimuli, coupled with their highly complex morphology [42] and the tremendous amount of cytoplasm present in the axonal compartment; indeed, RGCs have ~95% of their cytoplasm confined within their axons in the optic nerve [43]. Molecular chaperones are known to be required to maintain

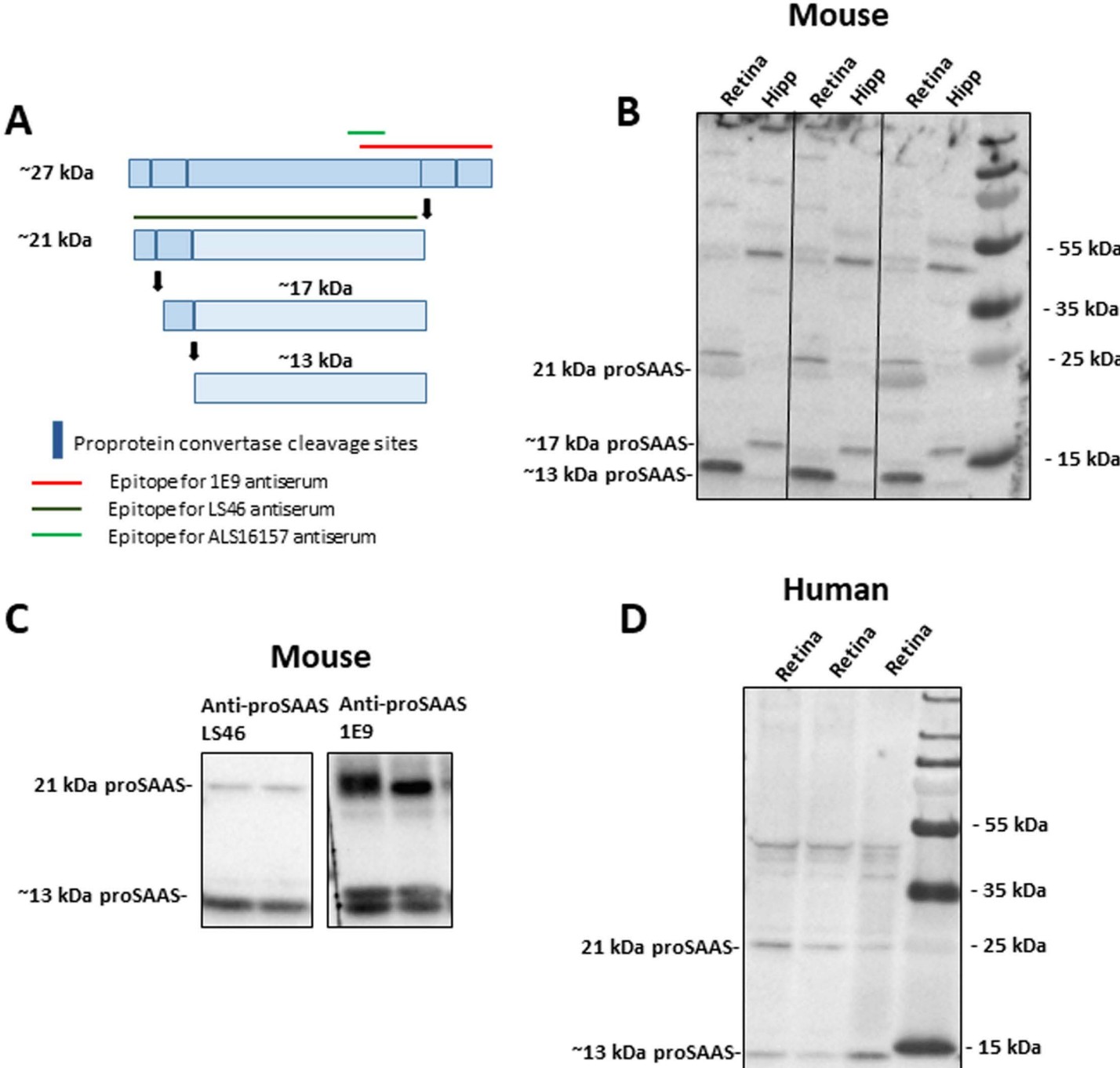

**Fig 6. Retinal proSAAS-immunoreactive species differ from those found in hippocampus. Panel A**: ProSAAS processing diagram depicting paired basic proprotein convertase cleavage sites (*blue vertical lines*); epitopes for the three proSAAS antisera used in this study (*red, dark green, and light green horizontal lines*); and theoretical sizes of each proSAAS fragment derived from intact 27 kDa proSAAS by proprotein convertase cleavage. **Panel B**: Western blotting of mouse retinal and hippocampal (*hipp*) acid extracts using LS46 antiserum. **Panel C:** Western blotting of retinal lysates using proSAAS antibodies LS46 and 1E9. Each lane represents pooled retinal or hippocampal extracts taken from an individual animal. **Panel D**: Western blotting of a human retinal extract (*retina*) using LS46 antiserum.

neuronal proteostasis and ensure neuronal health; this includes retinal neurons, which contain high levels of common neuronal chaperones [1,2]. For example, HSP90 is a cytoplasmic chaperone expressed at exceptionally high levels in RGCs; the majority of this protein is utilized in intra-axonal function [44]. However, the contribution of secretory protein chaperones to retinal cell biology and pathology has been relatively unexplored. Here, we investigated the expression and physical presence of the small secretory chaperone proSAAS in retinal tissues.

Examination of large RNAseq datasets from retinal samples shows similar distributions of proSAAS, clusterin, and αcrystallin-β transcripts between humans and mice, with mouse, though not human, retina exhibiting proSAAS expression levels roughly similar to clusterin. Biological reasons for these species differences have yet to be elucidated. Previous studies show that clusterin and αcrystallin-β exert protective effects in the retina [45,46], lending support to the hypothesis that proSAAS may function in a similar protective capacity in the retina.

Our review of disease-associated retinal RNAseq and proteomics studies also supports a potential positive association between proSAAS expression and retinal pathologies. Retinal proSAAS expression is significantly increased in three different disease states: glaucoma, retinitis pigmentosa, and age-related macular degeneration. The fact that proSAAS is a secreted protein suggests that it might play an important role in the extracellular milieu surrounding the RGC. Indeed, Ruzafa *et al.* [47] used an unbiased proteomics approach to examine candidate proteins secreted by Muller glia (a form of astrocyte) that protect ganglion cells; proSAAS was one of the top hits identified in this screen. However, demonstration of a causal protective relationship requires the use of recombinant proSAAS expression; these studies have not yet been performed.

The retinal transcriptomics meta-analyses presented here confirms that proSAAS is expressed at relatively high levels in retinal neurons, particularly in RGCs and horizontal cells. Retinal expression of proSAAS is further supported by Western blotting of retinal extracts, showing typical (21 kDa) and atypical (13 kDa) molecular mass forms, seen with two different proSAAS antisera. Our immunofluorescence data also experimentally confirm the majority of the meta-analysis findings, providing additional support for the idea that proSAAS might contribute to retinal proteostasis. However, some discrepancies between immunofluorescence and transcriptomics data are apparent. For example, immunoreactive proSAAS co-localizes to a greater extent with glial and bipolar cells than would be expected from the transcriptomic data. Differences between RNASeq and immunofluorescence experiments might arise in several ways. RNASeq studies do not reflect translational or post-translational regulatory mechanisms, such as miRNA regulation, ribosomal entry, post-translational modification, trafficking, and proteolysis.

Interestingly, both our immunofluorescence work and Broad Institute transcriptomics studies support the idea that proSAAS is also expressed in retinal astrocytes, albeit at a lower level than neurons [24]. Interestingly, a recent transcriptomics study showed that retinal astrocyte proSAAS expression increases as a function of aging [48].

The role of proSAAS in M1 RGC function potentially extends to circadian rhythm. Melanopsin-containing M1 RGCs convey circadian rhythm information to the suprachiasmatic nucleus [33,49]. Previous research has shown a role for a small proSAAS-derived peptide ("little SAAS") in mediating circadian rhythms within the superchiasmatic nucleus [41]. In agreement, proSAAS knockout mice display altered circadian rhythm entrainment [41,50,51]. Increased proSAAS transcript expression was also found in mouse retina following time shifting ["jet lag" experiment; [52]]. Lastly, proSAAS was also identified in synaptic proteomes associated with circadian rhythms in human samples [53]. Collectively, these reports, taken along with transcriptomics identification of enhanced proSAAS-ir in M1 RGCs both in the Broad study [30] as well as in another study [24] suggest an integrated role for proSAAS in circadian rhythm signaling across the eye and brain. However, because our immunofluorescence data do not support the enrichment of proSAAS in M1 RGCs seen in transcriptomics studies, additional experiments are required to validate this finding. Differences between mRNA and protein localizations could potentially account for these discrepancies.

Lastly, we show here that proSAAS expression is particularly rich in the synaptic (plexiform) layers of the retina and within processes between RGCs and amacrine cells. ProSAAS is a potent anti-aggregant chaperone [27,28] and

synaptically-localized hippocampal proSAAS has been previously shown to be both dynamically regulated [10,53] and released from hippocampal synapses in a primary cell system [11]. We hypothesize that proSAAS may also function as a chaperone at retinal synapses, contributing to synaptic and/or extracellular retinal proteostasis. A detailed analysis of proSAAS expression in the human retina in various retinal proteostatic diseases will provide much-needed information on the potential role of proSAAS as a retinal chaperone.

## Supporting information

**S1 File. SuppFiles.**
(PDF)

## Author contributions

**Conceptualization:** Nicholas Schaffer, Steven L. Bernstein, Iris Lindberg.

**Data curation:** Nicholas Schaffer, Samira Mitias, Yan Guo, Steven L. Bernstein.

**Formal analysis:** Nicholas Schaffer.

**Funding acquisition:** Steven L. Bernstein, Iris Lindberg.

**Investigation:** Nicholas Schaffer, Yan Guo, Samira Mitias.

**Project administration:** Steven L. Bernstein, Iris Lindberg.

**Resources:** Yan Guo, Steven L. Bernstein.

**Software:** Nicholas Schaffer.

**Supervision:** Steven L. Bernstein, Iris Lindberg.

**Visualization:** Nicholas Schaffer.

**Writing – original draft:** Iris Lindberg.

**Writing – review & editing:** Steven L. Bernstein, Iris Lindberg.

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
