## [Decision Letter · Decision Letter 0]

5 Jan 2025

PONE-D-24-48169The neuronal chaperone proSAAS is highly expressed in retinal neuronsPLOS ONE

Dear Dr. Lindberg,

Thank you for submitting your manuscript to PLOS ONE. After careful consideration, we feel that it has merit but does not fully meet PLOS ONE’s publication criteria as it currently stands. Therefore, we invite you to submit a revised version of the manuscript that addresses the points raised during the review process.

**ACADEMIC EDITOR: ** After careful consideration by 2 Reviewers and an Academic Editor, the Reviewers had a differing opinion on the merits of the submission. Accordingly, all of the critiques of the Reviewers, especially Reviewer #2, must be addressed in detail in a revision to determine publication status. If you are prepared to undertake the work required, I would be pleased to reconsider my decision, but revision of the original submission without directly addressing the critiques of the Reviewers (especially Reviewer #2) does not guarantee acceptance for publication in PLOS ONE. If the authors do not feel that the queries can be addressed, please consider submitting to another publication medium. A revised submission will be sent out for re-review. The authors are urged to have the manuscript given a hard copyedit for syntax and grammar.

We look forward to receiving your revised manuscript.

Kind regards,

Stephen D. Ginsberg, Ph.D.

Section Editor

PLOS ONE

Journal Requirements:

1. Please ensure that your manuscript meets PLOS ONE's style requirements, including those for file naming. The PLOS ONE style templates can be found at https://journals.plos.org/plosone/s/file?id=wjVg/PLOSOne_formatting_sample_main_body.pdf and https://journals.plos.org/plosone/s/file?id=ba62/PLOSOne_formatting_sample_title_authors_affiliations.pdf.

2. Please amend either the title on the online submission form (via Edit Submission) or the title in the manuscript so that they are identical.

[NIA AG062222  to IL

R01 EY032519  to SLB]. 

Please include this amended Role of Funder statement in your cover letter; we will change the online submission form on your behalf."

5. Please expand the acronym “NIA” (as indicated in your financial disclosure) so that it states the name of your funders in full. This information should be included in your cover letter; we will change the online submission form on your behalf.

6. In the online submission form, you indicated that [The data underlying the results presented in the study are available from I. Lindberg (ilindberg@som.umaryland.edu)].

7. PLOS ONE now requires that authors provide the original uncropped and unadjusted images underlying all blot or gel results reported in a submission’s figures or Supporting Information files. This policy and the journal’s other requirements for blot/gel reporting and figure preparation are described in detail at https://journals.plos.org/plosone/s/figures#loc-blot-and-gel-reporting-requirements and https://journals.plos.org/plosone/s/figures#loc-preparing-figures-from-image-files. When you submit your revised manuscript, please ensure that your figures adhere fully to these guidelines and provide the original underlying images for all blot or gel data reported in your submission. See the following link for instructions on providing the original image data: https://journals.plos.org/plosone/s/figures#loc-original-images-for-blots-and-gels.  

Reviewers' comments:

Reviewer's Responses to Questions

**Comments to the Author**

1. Is the manuscript technically sound, and do the data support the conclusions?

Reviewer #1: Yes

Reviewer #2: No

2. Has the statistical analysis been performed appropriately and rigorously? 

Reviewer #1: Yes

Reviewer #2: No

3. Have the authors made all data underlying the findings in their manuscript fully available?

Reviewer #1: Yes

Reviewer #2: Yes

4. Is the manuscript presented in an intelligible fashion and written in standard English?

Reviewer #1: Yes

Reviewer #2: Yes

5. Review Comments to the Author

Reviewer #1: This article by Schaffer et al. describes the expression pattern of the proSAAS chaperone in the mouse retina, focusing on RGCs.

The manuscript is well written and the experiments are well performed. The conclusions are consistent with the results.

There are only a few comments,

Brn3a+ RGCs are vision-forming RGCs and do not express melanopsin. Conversely, non-vision-forming RGCs express melanopsin (only M1-M3 subtypes express this protein at immunodetectable levels) but do not express Brn3a (for a detailed review of pan-RGC markers see PMID: 36594396). They should clarify both points in their manuscript.

As pNFH is highly expressed in RGC axons, immunodetection of this neurofilament with proSAAS, which appears to be axonal, may be more sensitive than biii tubulin.

The species of the sample should be clearly indicated in the figure legends.

Finally, although not compulsory, their RGD data would improve if they analyse proSASS expression on RGCs in mouse flatmounts

Note: The 5xFAD mouse strain is included in the methods but not in the results.

Reviewer #2: Review report outline

Research summary and overall impression

Summary of manuscript claims and overall evaluation.

The genesis of study by Schaffer et al. was to examine the protein expression levels and locations of the neuronal chaperone, proSAAS, an important chaperone for maintaining intra- and extracellular protein homeostasis. To this end, an initial meta-analysis on published single-cell retinal transcriptomes in humans and mice were conducted to assess retinal transcript levels and explore cell-specific expression patterns. The merged meta-analysis results indicated that retinal proSAAS gene expression was low-abundant and confined to retinal ganglion cells (epically photo-intrinsic retinal ganglion cells), amacrine cells, and horizontal cells. Complementary, retinal protein expression levels of proSAAS were investigated in murine (IHC of fixed retinal cross-section, Western blotting of retinal protein extracts) and human (retinal extracts) using antibodies directed against different isoforms of proSAAS and different retinal cell types. Based on the findings, Shaffer et al. claim to demonstrate that proSAAS is located primarily in the retinal nerve fibre, the ganglion cell, and the inner and outer plexiform layers.

The authors argued well for the relevancy of this study while structuring the manuscript in a concise manner making it easy for the reader to follow along. However, based on the methods and experimental findings in the manuscript and the supplementary data, there may be considerable risk of false claims and aberrant conclusions due to several factors compromising the interval validity of the study findings. In my opinion, these concerns needs to be adequately addressed to support several of the essential statements outlined in the manuscript.

Recommended course of action: Reject.

Several critical methodological issues needs to be addressed by the authors in order to adequately verify their results. The most important is the specificity of their utilized proSAAS antibodies. Other concerns is stated below.

Concerns

- In the methods and materials section associated with the “Animals” paragraph, the author states that both wild-type and 5xFAD mice for the ICC and western blot procedures. Please state exactly how many of each phenotype that have been used as well as other associated age and gender. Furthermore, is there a reason for pooling the retina and hippocampi of different phenotypic mice? Thence, it can’t properly be investigated if proSAAS is expressed in the retinas of both WT and Alzheimer’s mice (5x FAD) or in just one of the phenotypes. Hence, is proSAAS expressed in the “healthy” retina? Is proSAAS upregulated in the 5xFAD mouse retinas to counteract neurodegeneration? Is proSAAS expression diminished in 5xFAD murine retinas, suggesting that the loss of this proteostatic chaperone is implicated in the Alzheimer’s retinal pathophysiology? Moreover, despite storing the retinas and hippocampi on ice, a post-mortem interval of 1-6 h before progressing to tissue fixation or protein harvesting may substantially enhances the risk of the elicted results being influenced by neurodegeneration.

- In the methods section associated with the “Human tissue”. Please state the number and phenotypic characteristics of the donor patients. Is these healthy, young patients or diseased patients with local and systemic conditions affection the retina and/or optic nerve? Furthermore, what is the post-mortem time(s) before the retinas have been stored? How exactly have the retinas been cryopreserved? Furthermore, is there a reason why human retinas only have been used Western blotting, while IHC-analyses (which were performed on the murine retinas) have been left out?

- In the method section associated with the “meta-analysis” paragraph. On which criteria did you select the 2738 human retina samples? What log-transformation have been used? What is your scientific reasoning for removing outliers and zero-values from your analyses? Does inclusion of these values in your analysis affect the conclusion?

- Considering Figure 1A: I guess the y-axis should be proportion of samples? You transcriptomic quantity distributions are non-normally distributed. it would more appropriate to report median and interquartile ranges.

- Consider figure 1b: I guees the “gray quadrant” means “not detected”? please state the meaning? Furthermore, there is a huge interstudy difference in the normalized expression of proSAAS, which especially is low for Orozco and Wang. Could you identify any reasons for this? Could it be relevant to include in your discussion? Results of figure 1C could likewise be included in this discussion. Is there any reason why any statistical comparisons across the different cell-types within each study have not been performed in any of figure 1B and 1C?

- Considering figure 1D: While transcriptomic studies may indicate disturbances at the protein and functional levels, it is merely a surrogate. Hence, the proteomics studies is the only relevant studies to include in this figure if you want to argue that proteostasic dysregulation of proSAAS is present in several retinopathies. In that regard, only the glaucoma study by Mirzaei et al detects a statistically significant fold-increase of proSAAS. What kind of phenotypic glaucoma patients is this? At what disease stage(s)? Could be relevant in your discussion.

- A general concern of mine is the specificity of the antibodies used (LS46-serum derived and the commercial 1E9). It is good practice that non-primary negative controls have been used to assess the unspecific labelling patterns of the secondary antibodies used. However, this doesn’t take into account the specificity of your primary antibodies, which I fear may be low based on the results provided in Figure 2, 3, 4, 5. Could you provide evidence of the specificity of your antibodies against the different proSAAS isoforms, i.e., use siRNA for retinal knockout as a negative control? Determine the amino acid sequence of the protein(s) in the 13, 17, and 21 kDa bands derived from the retinal protein extracts to verify protein nature? Complement your analysis with FISH of proSASS-mRNA locational expression? MERFISH may be a good technology and allows for concomitant protein epitope labelling?

- A general comment to all of you ICC pictures. In general, I do believe that pictures could be of higher quality and magnitude. It is difficult to generate an impression at the cellular level.

- In figure 2, 3 and 4: Substantial background corrected fluorescens in the INL that almost seem identical to the IPL and GCL in all ICC figures. This somehow contrasts with the findings in Figure 1B showing none-to-mild translation of proSAAS in bipolar cells.

- In figure 2A and B: As you have based your statistical analysis on 22 slices stained with the LS46-antibdody and 6 with the 1e9 antibody, it would be appropriate to show representative pictures of both stainings. How have the retinal slices been selected for analysis? At random? Based on specific location from the optic nerve head? Why not conduct two separate analyses stratified on the antibodies used to see if the fluorescens patterns are the same or depend on the specific antibodies used? Why this asymmetric usage of antibodies-stained in your analyses i.e., 22 vs 6? Could it be an idea to supplement these data with a) a positive control i.e., hippocampal sections as you do in the western blots to assess whether the fluorescens signal likewise confines to the axons and synaptic regions, and b) proper negative controls as I previously mentioned.

- It is confusion that you use several synonyms for the same antibody i.e., Abcepta ALS16157 and 1E9. Please use consistent terminology.

- Regarding figure 3 and 4: Which ProSAAS antibody have been used?

- Figure 4B-D: Please provide full-pictures of the Western blots in order to give the reader an impression of the overall antibody specificity – especially for figure 4C. Why have there not been any pictures provided of western blots using the 1E9 antibodies in figure 4D?

- The authors claims in the abstract as well as in the introduction that the retina consists of 8 layers. I guess this should be corrected to 10 according to consensus view on the retinal anatomy. Otherwise please argue why.

- In section one of the introduction the reader is introduced to the term “proteostasis dysregulation”. Please define accordingly.

- In section one of the introduction the reader is introduced to the neuroretina, please define accordingly.

- In section two of the introduction the authors makes the following claim “Interestingly, while beta amyloid plaques are present in the aging brain and are especially prominent in Alzheimer’s disease (AD), there is a lack of amyloid plaque development in the retina, despite high concentrations of soluble amyloid precursor protein”. Please cite accordingly for this claim. However, I do not believe that the most recent literature on Alzheimer’s manifestations in the retina support this claim. As such, β-amyloid beta plaques have been demonstrated in the retinas of diseased patients with none to severe symptoms, and thus may find implications in the clinic as a biomarker. Please see the following review: Wang L, Mao X. Role of Retinal Amyloid-β in Neurodegenerative Diseases: Overlapping Mechanisms and Emerging Clinical Applications. Int J Mol Sci. 2021 Feb 26;22(5):2360. doi: 10.3390/ijms22052360. PMID: 33653000; PMCID: PMC7956232.

- In section three of the introduction the authors makes the following statement “While numerous studies support the theory that proSAAS is a proteostatic stress-reducing chaperone, few studies have examined retinal proSAAS expression”. Please cite the studies mentioned and inform the reader what your study contributes to the literature that these other studies don’t.

- Study limitations not discussed in the “discussion” section

- Figure 4A: A two-way ANOVA is not the correct statistical test to use. Given that you are comparing the CTF (one dependent variable) across more than two (unpaired) groups defined by a single factor (epitope labelling), a One-Way ANOVA with posthoc comparison and correction should be employed depending on the parametric distribution of you retina. Moreover, what does the vertical bars represent? In figure 4A, please set the y-scale to start at 0. Have there also been conducted a Two-Way Anova in figure 4B?

- In Figure 2B: What kind of unpaired t-test has been used? Why haven’t there been performed statistical comparisons across other groups?

6. PLOS authors have the option to publish the peer review history of their article (what does this mean? ). If published, this will include your full peer review and any attached files.

**Do you want your identity to be public for this peer review?** For information about this choice, including consent withdrawal, please see our Privacy Policy .

Reviewer #1: **Yes: ** MARTA AGUDO-BARRIUSO

Reviewer #2: **Yes: ** Alexander on Spreckelsen

---

## [Author Response · Author response to Decision Letter 0]

5 Mar 2025

Please see the Cover letter and the Rebuttal Letter.

---

## [Decision Letter · Decision Letter 1]

12 Mar 2025

The neuronal chaperone proSAAS is highly expressed in the retina

PONE-D-24-48169R1

Dear Dr. Lindberg,

We’re pleased to inform you that your manuscript has been judged scientifically suitable for publication and will be formally accepted for publication once it meets all outstanding technical requirements.

Kind regards,

Stephen D. Ginsberg, Ph.D.

Section Editor

PLOS ONE

**Comments to the Author**

1. If the authors have adequately addressed your comments raised in a previous round of review and you feel that this manuscript is now acceptable for publication, you may indicate that here to bypass the “Comments to the Author” section, enter your conflict of interest statement in the “Confidential to Editor” section, and submit your "Accept" recommendation.

Reviewer #1: All comments have been addressed

2. Is the manuscript technically sound, and do the data support the conclusions?

Reviewer #1: Yes

3. Has the statistical analysis been performed appropriately and rigorously? 

Reviewer #1: Yes

4. Have the authors made all data underlying the findings in their manuscript fully available?

Reviewer #1: Yes

5. Is the manuscript presented in an intelligible fashion and written in standard English?

Reviewer #1: Yes

6. Review Comments to the Author

Reviewer #1: The authors have addressed all the concerns raised, and the article has undergone significant improvement as a result.

7. PLOS authors have the option to publish the peer review history of their article (what does this mean? ). If published, this will include your full peer review and any attached files.

**Do you want your identity to be public for this peer review?** For information about this choice, including consent withdrawal, please see our Privacy Policy .

Reviewer #1: No

---

## [Editor Report · Acceptance letter]

PONE-D-24-48169R1

PLOS ONE

Dear Dr. Lindberg,

I'm pleased to inform you that your manuscript has been deemed suitable for publication in PLOS ONE. Congratulations! Your manuscript is now being handed over to our production team.

Kind regards,

on behalf of

Dr. Stephen D. Ginsberg

Section Editor

PLOS ONE